# Suitability of Laser Engineered Net Shaping Technology for Inconel 625 Based Parts Repair Process

**DOI:** 10.3390/ma14237302

**Published:** 2021-11-29

**Authors:** Izabela Barwinska, Mateusz Kopec, Magdalena Łazińska, Adam Brodecki, Tomasz Durejko, Zbigniew L. Kowalewski

**Affiliations:** 1Institute of Fundamental Technological Research, Polish Academy of Sciences, Pawińskiego 5B, 02-106 Warsaw, Poland; mkopec@ippt.pan.pl (M.K.); abrodec@ippt.pan.pl (A.B.); zkowalew@ippt.pan.pl (Z.L.K.); 2Department of Mechanical Engineering, Imperial College London, London SW7 2AZ, UK; 3Faculty of Advanced Technologies and Chemistry, Military University of Technology, Sylwestra Kaliskiego 2, 00-908 Warsaw, Poland; magdalena.lazinska@wat.edu.pl (M.Ł.); tomasz.durejko@wat.edu.pl (T.D.)

**Keywords:** LENS technology, Inconel alloys, repair process, additive manufacturing

## Abstract

In this paper, the Inconel 625 laser clads characterized by microstructural homogeneity due to the application of the Laser Engineered Net Shaping (LENS, Optomec, Albuquerque, NM, USA) technology were studied in detail. The optimized LENS process parameters (laser power of 550 W, powder flow rate of 19.9 g/min, and heating of the substrate to 300 °C) enabled to deposit defect-free laser cladding. Additionally, the laser clad was applied in at least three layers on the repairing place. The deposited laser clads were characterized by slightly higher mechanical properties in comparison to the Inconel 625 substrate material. Microscopic observations and X-ray Tomography (XRT, Nikon Corporation, Tokyo, Japan) confirmed, that the substrate and cladding interface zone exhibited a defect-free structure. Mechanical properties and flexural strength of the laser cladding were examined using microhardness and three-point bending tests. It was concluded, that the LENS technology could be successfully applied for the repair since a similar strain distribution was found after Digital Image Correlation measurements during three-point bending tests.

## 1. Introduction

Inconel 625 is a multicomponent nickel-based alloy with unique properties, that enable its application in fast-growing industries like aviation and cosmonautics. This alloy is characterized by high heat resistance, and thus, it could be used in aggressive environments. The heat resistance of Inconel 625 is attributed to the content of refractory alloying addition including molybdenum and niobium [1]. Additionally, it has a high strength over a wide range of temperatures up to 1170 °C. This factor makes Inconel 625 a promising candidate for machine parts working under extreme conditions, e.g., jet engines and gas turbine blades [1,2,3].

It should be mentioned, however, that an extended operational time at high temperatures often leads to damage of the engine’s crucial parts. Moreover, manufacturing faults made at a component’s design stage may lead to rapid crack initiation and subsequent substantial damage development as the defected part is more susceptible to excessive wear. Among possible ways to reduce the high cost of a new part made of the Inconel alloys, one can indicate a repairing. The repair process is beneficial for a reduction of the general system maintenance cost, and more importantly, it is environmentally-friendly, because it minimizes the need for damaged component utilization [4]. Moreover, it also enables to save time. It has been found, that thanks to this process the costs of a new part can be reduced even by 45% [5]. The most important issue related to the repair process is recreating the geometry and reconstructing or improving the initial properties of the damaged part. It could be performed by several techniques depending on the damaging nature. One of the conventionally used processes of repair is a hard facing. This process consists of the local heating to the melting point of the element surface and additional parts in order to permanently connect both of them. Another example of the repair process is thermal spraying, in which fully molten or partially melted materials in the form of a droplet, are sprayed on the surface. The process is followed by rapid cooling and solidification [6,7]. It should be highlighted, that the Gas Tungsten Arc (GTA) welding process is typically used for Inconel components repair. In this process, the non-consumable tungsten electrode is used to obtain an electric arc. The arc is required to melt the filler metal, which is added either manually or automatically. The tungsten electrode and the molten metal are shielded with an inert gas during the process. The inert gas shielding protects the weld pool area against the atmosphere and possible contamination [8,9]. GTA provides high-quality, precise, and defect-free welds. The GTA welding equipment is affordable and could be used for onsite welding works. However, it is not recommended to use such a process for the repair of the Inconel 718 alloy, since the high heat generated during this process promotes segregation of the brittle phases and grain growth [10]. As another possible method of the nickel-based alloys repairing one can indicate plasma and microplasma transferred arc processes (PTAW), being a modified version of GTA. The fundamental difference is that PTAW involves constriction of the arc, which increases the amount of ionization (or plasma) during the process, causing an increase in the arc temperature [11]. Conventional methods of regenerative material usage have a number of disadvantages that often disqualify them from applications. However, the laser source-based methods offer an overcoming of the main issues related to the conventionally used methods [12] as well as enable deposition of the complex, even of high-entropy, coatings in order to enhance the mechanical properties of parts working under extreme conditions [13]. Hong et al. [14] used the laser metal deposition (LMD) process to produce an ultrafine TiC particle in order to reinforce Inconel 625 composite parts, which were characterized by a significantly high value of the ultimate tensile strength of 1077.3 MPa, yield strength of 659.3 MPa, and elongation of 20.7%. Weng et al. [15] presented a repair process approach by using the laser aided additive manufacturing with powder flow rate (LAAM) which was successfully applied to deposit the SS410 or Inconel 625 on SS416 substrate. It was shown, that deposited clad exhibited no obvious defects, and the interface samples exhibited comparative or slightly lower ultimate tensile strength in comparison to the SS416 substrate. One of the promising laser cladding methods is LENS. During this process, a focused laser beam, together with powder, that is fed into a high-vacuum chamber by an inert gas, forms a pool on the surface of the substrate material, resulting in the stable connection of materials. The LENS system enables both, repair of parts and surface modification by the application of a protective coating since the process is characterized by high accuracy and purity during laser cladding [16,17,18]. An important advantage of using this system from the metallurgical point of view is the narrower heat-affected zone than that usually obtained in the conventional methods. Other methods deliver large amounts of heat deeply into the material, while in the LENS process the high energy of the laser beam is focused in the relatively small area. In such a case, no overheating and subsequent material deformations, as well as stress concentrations, can be observed [19]. This advantage is very important during the repair of thin-walled components. The high power density of the laser also promotes the application of refractory materials and increases the speed of the process. The resulting laser clads are narrower than those in any other process, and as a consequence, the final part does not require either rough or finish machining. The additive nature of the LENS process makes it a more resource-efficient manufacturing technology since less waste is generated in comparison to the subtractive techniques [11,12,13,14,15,16,17,18,19,20]. Additionally, a less amount of the material could be used for the part repair, which is extremely important in terms of the application of nickel-based alloys.

Nickel alloys, despite their excellent resistance to high temperature, tend to form brittle intermetallic phases, which decrease the mechanical properties of the material [21]. Moreover, the high production cost of parts made of the Inconel alloys demands new techniques, that enable the fabrication of complex geometries [22] or repair the broken part without the necessity of their replacement. Therefore, the main aim of this work was to assess the suitability of the Laser Engineering Net Shape Technology to repair parts made of the Inconel 625 nickel-based superalloy deposited using the optimized process parameters. Moreover, since the LENS technology reduces an area of heat-affected-zone and does not change the physical characteristics of the deposited material and the substrate, such aspects of this technique were also studied with regard to damaged parts repair.

## 2. Materials and Methods

The chemical composition, the morphology of powder particles, and EDS maps were captured by using FEI Quanta scanning electron microscope with Energy Dispersive Spectroscopy (EDS, FEI, Hillsboro, OR, USA) (Figure 1, Table 1). The material for tests was delivered in form of powder of the Inconel 625. A chemical composition of the nickel-based superalloy was presented in Table 1. The specimens for microstructural observations were etched by using oxalic (90 mL of water +10 mL oxalic acid).

Based on the particle size analysis, it was determined, that the average size of the powder was 70 μm (Figure 2). The LENS process parameters were optimized using the LENS 850-R system (Optomec, Albuquerque, NM, USA) on the model specimens of substrate Inconel 625. The technological route includes the selection of the LENS parameters on the substrate material and subsequent application of such parameters for 1 mm depth pocket repair in order to optimize them. The optimized parameters are used to fill the pocket with a finishing allowance of 0.1–0.3 mm and a transition zone of high metallurgical quality using the continuous-wave operation mode of the laser. Finally, the repair process was performed on rectangular-shaped pockets from which the bending specimens were cut. Additionally, the optimized LENS process parameters were used to prepare additive manufactured specimens with the same geometry as repaired and wrought ones in order to compare their mechanical response. During the LENS repair process, the level of oxygen and water vapor was maintained at less than 7 ppm by using the argon protective atmosphere.

After process completion, the welding results were analyzed along the cross-section to check the quality of the connections between the additional material and substrate. The microstructural observations were analyzed by using an optical microscope Nikon MA220 (Nikon Corporation, Tokyo, Japan) and Nikon X-Tek Xt H225 Micro-Ct tomograph (Nikon Corporation, Tokyo, Japan). Tomographic scanning was carried out using a micro focal X-ray source with 120 kV energy and a focal spot size ~3 µm. The number of projections around 360° was equal to 1000. The microhardness of cladding was determined on a ZWICK hardness tester (Materialprüfung, Ulm, Germany). It was measured every 0.1 mm starting from the edge of the cladding up to its core and repeated in five different cross-sections of the repaired specimen. The three-point bending test was carried out by means of a servo-hydraulic Instron 1343 testing machine (Instron, Norwood, MA, USA) to assess the effectiveness of the LENS technology for material repair. In this research, the rectangular-shaped specimens of 70 × 5 × 5 mm made of the as-received, additively manufactured, and repaired Inconel 625 were tested at room temperature to compare their mechanical responses depending on the material state. The mechanical testing was executed under displacement control at the strain rate of 2 × 10^−3^ s^−1^. Based on the experimental results, bending force and bending strength were determined. The bending tests were monitored by DIC Aramis 12M (GOM mbH, Braunschweig, Germany) equipped with lenses of a total focal length of 75 mm and calibration settings appropriate to the measuring area equal to 170 × 156 mm. The calibration was performed before testing using a certified GOM calibration plate. General views of the experimental setup are presented in Figure 3. The bending zone of the specimens deformed was observed using scanning electron microscopy.

## 3. Results and Discussion

### 3.1. Optimization of Process Parameters

Optimization of the LENS process parameters could lead potentially to the development of AM process, which enables the Inconel 625 part repair. Moreover, the LENS process application seems to be beneficial because it gives a good opportunity to overcome the significant problem related to the high production cost of parts made of the Inconel alloys.

Preliminary technological tests of powder deposition on the Inconel 625 substrate were carried out using the Line Build Deposition module. That module enabled the determination of the process parameters in order to achieve the assumed thickness of the deposited clads (Figure 4). During the laser cladding by Line Build Deposition modulus, the test specimens were produced by keeping the process parameters constant at a laser power of 550 W, laser spot of 1.5 mm, and shielding and carrier gas flow rate of 20 L per minute (LPM) and 6 LPM, respectively. The stand-off distance was 13.7 mm. A parameter, that was changed during the test, was the powder flow rate. For each of the test variants considered, their thicknesses were measured to determine the powder flow rate that enables us to get a deposition thickness of 1 mm, approximately, in a single pass. This process was subsequently repeated, however, the powder was applied by means of the Teach and Learn module, in which the contour was built firstly, and then, the hatch was deposited (Figure 4. (6–13)). Based on the macrostructural observations, it was found that the deposited laser clads do not meet the geometry and quality requirements. It should be mentioned, that during contour deposition, too much powder was applied during the process, which in turn led to an uncontrolled increase of its thickness. Subsequent technological trials aimed to select proper process conditions, that enables obtaining a uniform thickness over the entire surface. It was decided to change the Laser On/Off Wait parameter, which specifies how long the delay between depositing each line of material should last. Additionally, it also excludes the extensive deposition of powder near the wall of the repaired pocket. The parameters used to optimize the LENS surfacing system were shown in Table 2. As the process parameters used during deposition did not enable to obtain a satisfactory quality of the clad, the temperature of the Inconel 625 substrate was increased up to 300 °C. Finally, the optimized parameters were found and shown in Table 3.

Cross-sections of the laser cladding specimen deposited on a 1 mm depth pocket using the set of parameters denoted as 13 in Table 2 were shown in Figure 5. The microscopic and tomographic observations revealed, that between the substrate and deposited material many defects can be found including local discontinuities in areas between substrate material and cladding (Figure 5a,d), porosity (Figure 5b), and unmelted particles (Figure 5c). These issues were related to the relatively low temperature of substrate material during the deposition process in which un-melted or partially melted powder particles could be formed at the interfacial regions between successive layers. Moreover, due to the large discontinuities observed in these regions the powder stream and laser beam were intensively scattered at the edges of the pocket. It should be mentioned, however, that the repair process performed on the same 1 mm depth pocket of a heated substrate up to 300 °C in the case of three layers improves the quality of the connection. The material obtained can be treated as a non-defected laser clad with a homogenous structure and specified thickness (Figure 6). The observations performed using the optical microscope (Figure 6a) on the cross-section of the deposited specimen revealed the permanent connection between clad and substrate material. Additional X-ray Tomography (XRT, Nikon Corporation, Tokyo, Japan) investigations carried out in the volume of the repaired specimen confirmed the correctness of the optimized process parameters. No discontinuities were found within the material.

Similar work was reported by Kumar et al. [23], where the Inconel 718 was applied to a substrate of the same material by laser metal deposition. The repair process was carried out using the following parameters: laser power 2350 W; powder flow rate 10 g/min; and laser scan speed 700 mm/min. It was found, that the optimized process parameters could be directly used to repair critical aero-engine components made of the Inconel 718. Petrat et al. [24] also presented a repair technology by using laser metal deposition during which a turbine made of the Inconel 718 was repaired. In this study, a TRUMPF TruDisk (TRUMPF, Ditzingen, Germany) 2.0 kW Yb:YAG laser equipped with 3 jet powder nozzle working under carrier gas Helium 5.0 with 4 L/min and protective gas Argon 5.0 with 10 L/min was used. The average size of the powder particles was equal to 67.5 µm.

Lugan et al. [25] reported Nd:YAG laser direct metal deposition technique for the 738LC nickel superalloy repair. In this work, a combination of parameters was selected to P/V (laser power/travel speed) of 24–30 J/mm and M/V (powder flow rate/travel speed) of 30–37 g/m. Such process parameters enabled obtaining a repaired material with 80–85% of the parent material properties for the high-temperature tensile properties at 850 °C and with 20% creep resistance stress reduction. Liu et al. [26] found, that the LENS is an effective method for repairing casting defects and misplaced machining holes in the Ni-base superalloys. The Inconel 718 tested in this research exhibited excellent repair weldability without visible defects except for some minor porosity observed in the material of repairs representative of deep through-holes.

### 3.2. Characterization of Laser Cladded Microstructure

The microstructure of the laser cladding of Inconel 625 deposited by means of the LENS technique was presented in Figure 7. The micrographs were captured near the interface between the substrate and deposited material using the optical microscope. A relatively narrow heat-affected zone of about 35.2 μm can be observed. In the areas where the laser started and finished the process, the heat-affected zone was slightly wider ranging from 50 to 70 µm (Figure 7). Similar observations of the heat-affected zone were performed by Liu et al. [26], who used the LENS Technology for the repair of Inconel 718 superalloy turbine components. It was found, that the LENS could be successfully applied for the repair process and prevent effectively the phase transformations on the border of the cladding and substrate material resulting from the extremely high temperature. Zhang et al. [27] performed the repair of the Inconel 718 components by means of laser additive manufacturing. In this study, the Inconel 718 was deposited on the substrate of the same material with a trapezoid grove by a fiber-coupled diode laser system (Laserline LDF400–2000, Mülheim-Kärlich, Germany) which was characterized by the wavelength of 980 nm and maximum laser power of 2 kW. The parameters used during the process were: laser power (P) of 900 W, scanning velocity (V) of 10 mm/s, powder-feeding rate (F) of 10 g/min, carrier gas flow rate of 15 L/h, and overlapping ratio (R) of 40%. The microscopic observation results show, that there is also a small heat-affected zone of approx. 50–80 µm and there were no pores and cracks in the repaired zone. Similar results were reported by Vemanaboina [28], where the Inconel 625 was welded using CO_2_ Laser Beam and was tested in order to determine selected mechanical and metallurgical properties. It was observed, that the laser power of both 3.0 kW and 3.3 kW and a welding speed of 1.0 m/min enables to obtain the best metallurgical quality of the weld.

Subsequently, the energy-dispersive X-ray spectroscopy (EDS) maps of the laser cladding on the substrate were elaborated. The EDS maps from the scanning microscope with the EDS detector enabled the assessment of the element distribution of the LENS cladding. The distribution of the based elements observed on the cross-section confirmed, that the LENS system lead to the slight segregation of the alloying elements on the border of cladding and substrate material. Niobium and molybdenum were observed on grain boundaries of cladding (Figure 8). According to Yang et al. [29], molybdenum could reduce the solubility of niobium in the dendrite arm and Laves phases. Additionally, the molybdenum addition transforms the Laves phase morphology and decreases the segregation zone around the Laves phase. Moreover, the research of Zhang et al. [27] presents, that the laser repair technology applied for the Inconel 718 alloy components led to elemental precipitation and the occurrence of the Laves phase. It could be concluded, that the process parameters proposed by the authors in a recent study enable a slight reduction of a phase transformation that occurred due to the high temperature resulting from the relatively high laser power of 900 W.

### 3.3. Microhardness Profile

The hardness distribution was measured from the edge of the laser clad in its cross-section (Figure 9). The dots represent the average value from 5 measurements while the maximum and minimum values of the error bars represent the maximum and minimum values of the hardness measurements, respectively. The microhardness of the laser clad zone was equal to 269 HV0.1 and decreases towards the surface layer of nickel-based superalloy to approx. 249 HV0.1. The hardness of the heat-affected zone was approx. 252 HV0.1. Abioye et al. [30] presented research devoted to deposition of the Inconel 625 on the steel to achieve its protection against corrosion. Deposition of the Inconel 625 wire on the steel specimen was carried out with the fiber laser (IPG Photonics, manufacture, city, country) operating at 1070 nm wavelength. The hardness value was checked during the test for both the substrate and the laser clad. The results confirmed that the hardness of the substrate was not altered significantly during the test. Moreover, Verdi et al. [31] investigated the mechanical properties of laser-applied Inconel 625 coatings with dimensions of 45 mm × 30 mm × 6 mm on the steel specimens. The deposition process was carried out using a High-Power Diode Laser (HPDL) with a wavelength of 940 nm and maximum output power of 1300 W. During the process, the laser power was 900 W, while other parameters such as scanning speed and supply filler material were 15 mm/s and 16.5 g/min, respectively. The results of the Vickers hardness measurements showed that the hardness of the coating is greater than that of the substrate by approximately 80–100 HV0.3.

### 3.4. Three-Point Bending Tests Supported by Digital Image Correlation Technique

The bending tests were carried out on three different specimens to compare the properties of the substrate material, additive manufactured material, and material with the LENS cladding (Figure 10, Table 4). It was found, that the specimen produced by the additive manufacturing was characterized by the greatest bending strength of 1535 ± 40 MPa. It was slightly lower for the specimens with laser cladding being 20 MPa less than the previous one. The lowest level of this parameter was achieved for the specimen made of the substrate material. It was 1460 ± 10 MPa. It should be highlighted, that similar strain distribution was observed in all types of the specimens tested regardless of the manufactured process (Figure 11). Therefore, it could be concluded, that the proposed parameters for the Inconel 625 repair using the LENS technology were selected properly.

Wei et al. [32] investigated the mechanical properties of the laser additive repaired Inconel 625. The main process parameters included the laser power of 1.4 kW, scanning speed of 400 mm/min, the laser spot diameter of 3 mm, and powder flow rate of 6 g/min. The repaired specimens were characterized by similar tensile strength and hardness as the as-received material proving, that the proposed methodology is suitable for the repair process. Sui et al. [33] also used the laser repaired method to assess the suitability of such method for the Inconel 718. The process parameters involved the laser power of 1.5–2.0 kW, scanning speed of 8–10 mm/s, the laser spot diameter of 2 mm, and powder flow rate of 5 g/min. The tensile tests performed in this study revealed, that the tensile properties of the repaired material were 90% of those of the as-received material.

In our research, an observation of the test specimens did not reveal the surface cracks. This indicates that laser deposition did not reduce the strength of the component. In order to identify deformation mechanisms, the side of each specimen was polished before the bending tests (Figure 12a). The scanning electron microscopy observations did not show cracks on the side of the substrate specimen (Figure 12b). On the other hand, the deformation of the LENS cladded specimen resulted in the exposure of the deposited zone (Figure 12c). However, no cracks were found between substrate and cladding. One can confirm, that the optimized repair process parameters led to the durable connection of the substrate material and deposited zone. Moreover, a slightly higher bending strength of the repaired specimen in comparison to substrate one was clearly demonstrated. Additionally, bending of the additive manufactured specimen identified the subsequent cladded layers (Figure 12d). Such type of the Inconel 625 was characterized by the highest bending strength. Some variations in the mechanical response of the Inconel 625 alloy in its different states resulted from their initial microstructures and were revealed after observations of the bending area. The repaired and additively manufactured Inconel 625 were characterized by the columnar structure (Figure 12c,d). Moreover, the laser-based technologies promote an occurrence of the Laves phases which are beneficial with regard to the mechanical properties of the Inconel alloys [34].

## 4. Conclusions

Optimization of the LENS parameters for the Inconel 625 nickel-based superalloy with respect to the repair process enabled obtaining a non-defected laser clad with the specified thickness of about 1 mm. The laser clad was characterized by a very good adherence to the substrate material and even improved mechanical properties. Moreover, the relatively thick heat-affected zone of about 35 µm was generated. The LENS technology with parameters carefully determined (laser power of 550 W, feed rate of 12 mm/s, and powder flow rate of 19.9 g/min) can be successfully used to repair the parts of nickel-based superalloy. It should be highlighted, that the higher temperature of the material substrate (300 °C) and relatively low power of laser reduced the significant phase transformations commonly found in the conventional repair processes.

## Figures and Tables

**Figure 1 materials-14-07302-f001:**
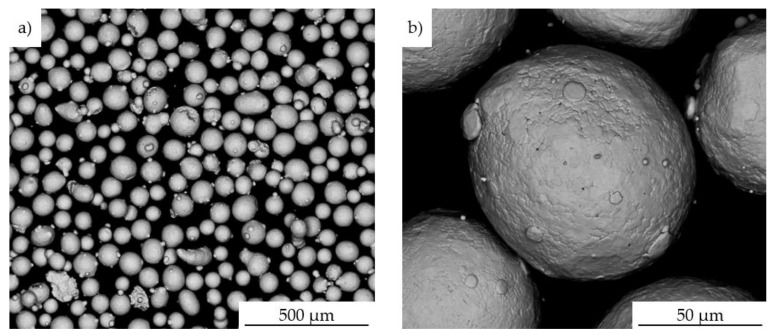
General view of the Inconel 625 batch powder (**a**); view of a single powder particle (**b**).

**Figure 2 materials-14-07302-f002:**
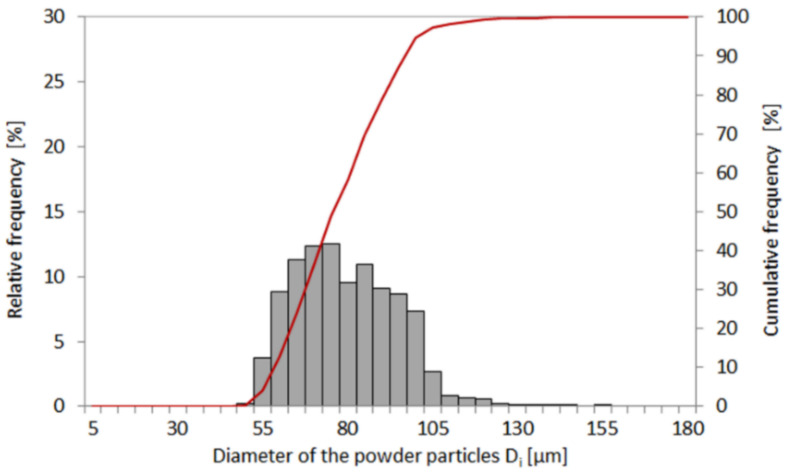
Particle size distribution of Inconel 625 powder (columns–relative frequency, line–cumulative frequency).

**Figure 3 materials-14-07302-f003:**
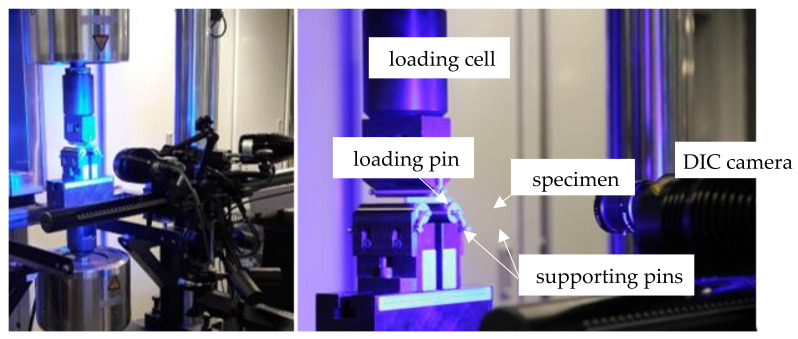
Experimental setup for the three-point bending test supported by DIC technique.

**Figure 4 materials-14-07302-f004:**
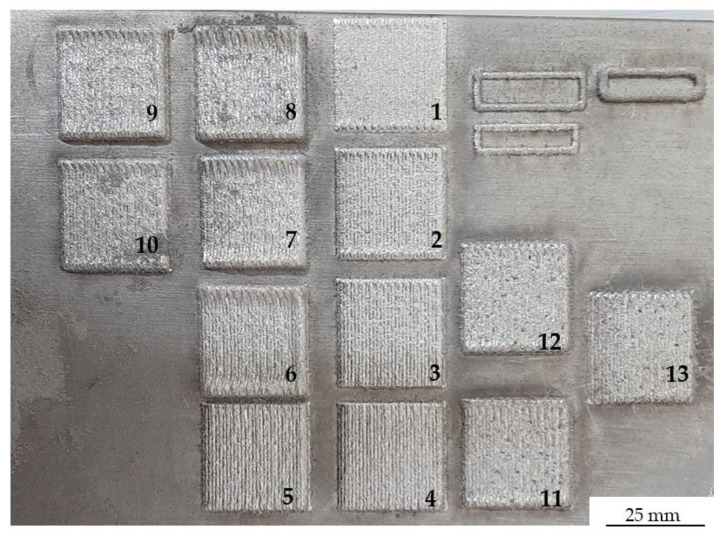
Effects of initial technological trials by using the LENS system and Inconel 625 powder: 1–5 deposited with the Line Build Deposition module, 6–13 deposited with the Teach and Learn module.

**Figure 5 materials-14-07302-f005:**
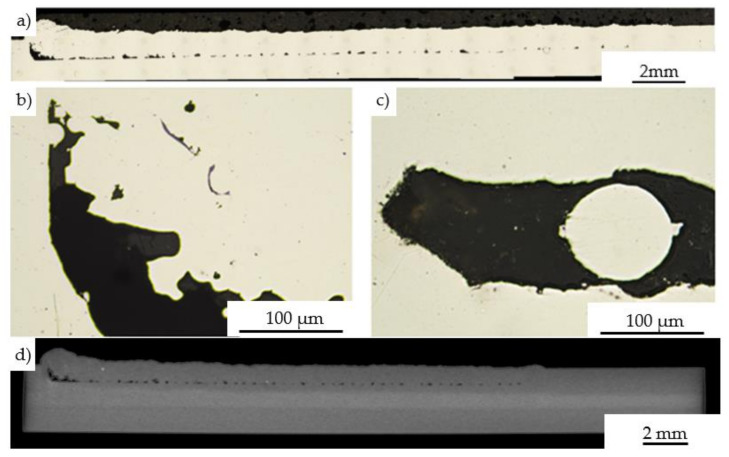
Microstructure of the longitudinal section without optimization of the application parameters (**a**); discontinuities found in between substrate material and cladding (**b**,**c**); tomographic longitudinal section of the cladding (**d**).

**Figure 6 materials-14-07302-f006:**
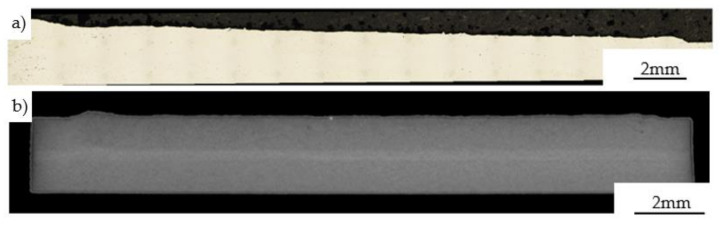
Microstructure of the longitudinal section with optimization of the application parameters (heated substrate, three layers across the thickness) (**a**); tomographic longitudinal section of the final cladding (**b**).

**Figure 7 materials-14-07302-f007:**
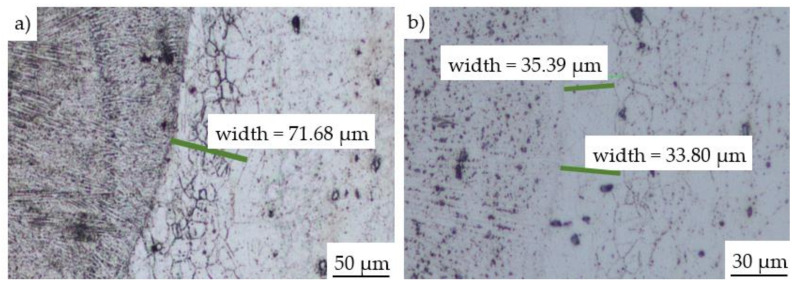
The cross-section of the Inconel 625 laser clad with visible heat affected zone near the deposition starting point (**a**), and 3 mm further (**b**).

**Figure 8 materials-14-07302-f008:**
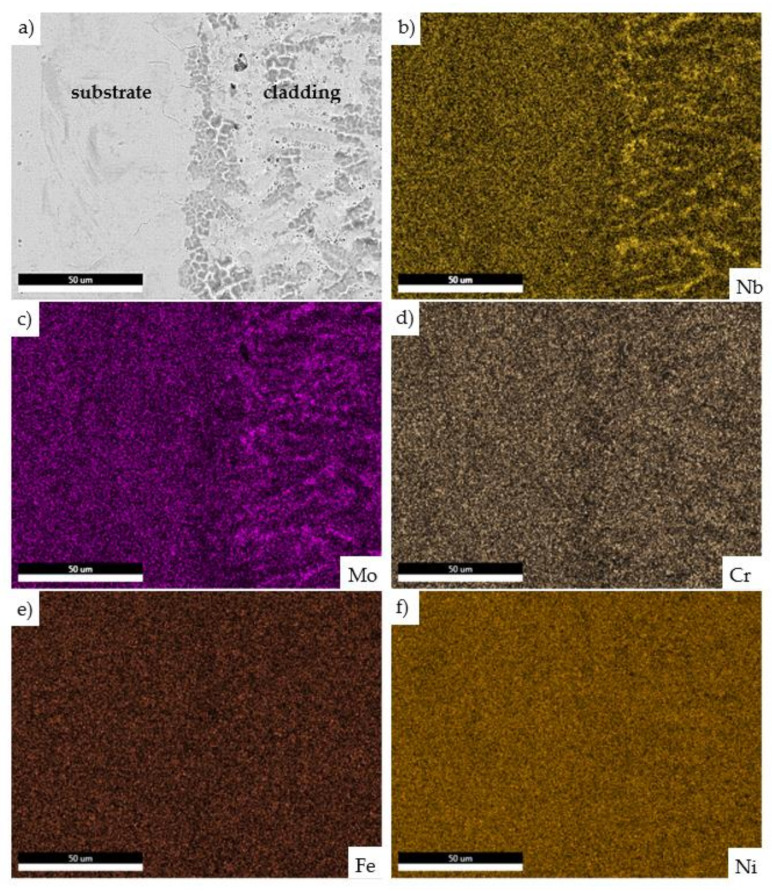
Connection of the cladding and substrate material (**a**); energy-dispersive X-ray spectroscopy (EDS) maps of niobium (**b**); molybdenum (**c**); chromium (**d**); iron (**e**); nickel (**f**) distribution.

**Figure 9 materials-14-07302-f009:**
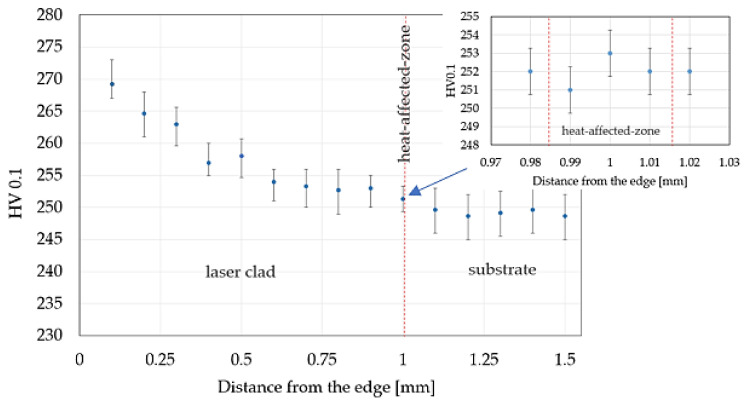
Hardness distribution captured in the cross-section of laser cladding.

**Figure 10 materials-14-07302-f010:**
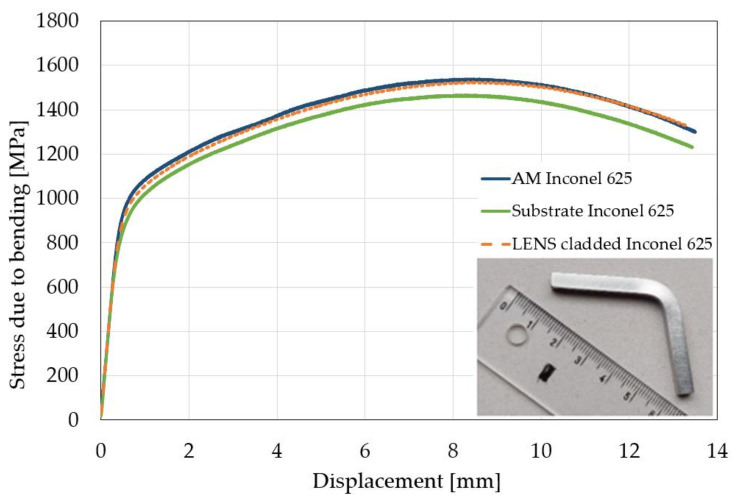
Characteristics of the stress due to bending versus displacement determined from the three-point bending tests.

**Figure 11 materials-14-07302-f011:**
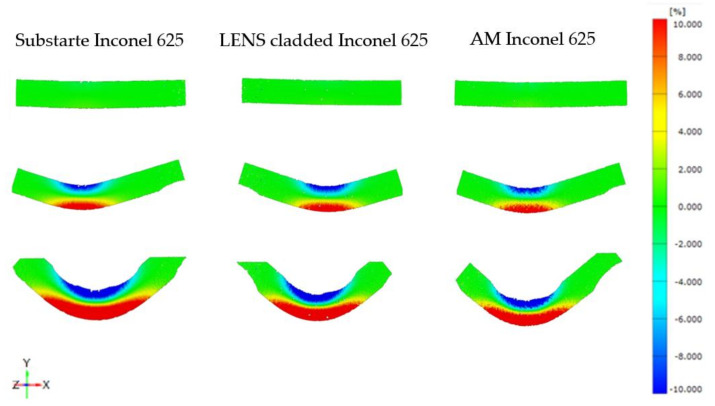
DIC strain distribution maps of the Inconel 625 in the as-received state, with additional clad and additively manufactured.

**Figure 12 materials-14-07302-f012:**
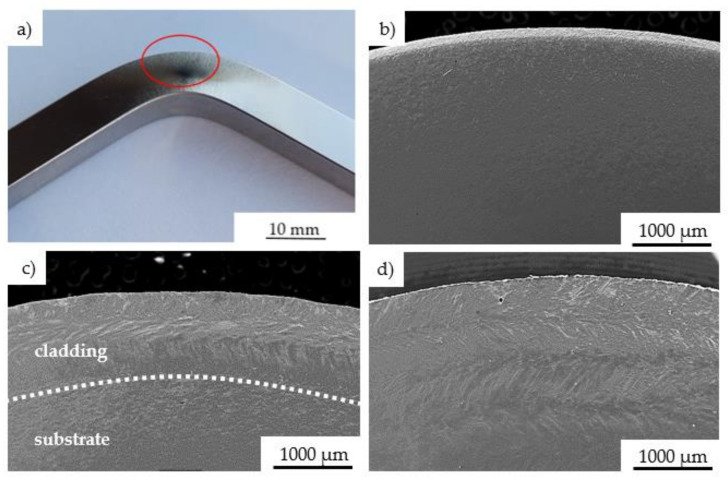
Surface of the specimen after bending test: area of microscopic observations (**a**); substrate Inconel 625 (**b**), LENS cladded Inconel 625 (**c**), AM Inconel 625 (**d**).

**Table 1 materials-14-07302-t001:** Chemical composition of the Inconel 625 nickel superalloy (%wt.).

Element	O	Fe	Ni	Al	Si	Zr	Nb	Mo	Cr	Mn
%wt.	2.18	5.10	55.87	0.38	0.34	0.48	3.83	9.03	22.53	0.27

**Table 2 materials-14-07302-t002:** Parameters used for the optimization of the LENS repair process.

No.	Laser Power [W]	Feed Rate [mm/s]	Powder Flow Rate [g/min]	Laser On/Off Wait [ms]	Substrate Temperature [°C]
1	550	10.5	9.9	-	23
2	550	10.5	13.2	-	23
3	550	10.5	16.6	-	23
4	550	10.5	18.3	-	23
5	550	10.5	19.9	-	23
6	550	9	19.9	1	23
7	550	9	19.9	1	23
8	550	9	19.9	1	23
9	550	9	19.9	50	23
10	550	9	19.9	100	23
11	550	10.5	19.9	200	23
12	550	12	19.9	400	23
13	550	12	19.9	400	23

**Table 3 materials-14-07302-t003:** Parameters used for the optimization of the LENS repair process.

Laser Power [W]	Feed rate [mm/s]	Powder Flow Rate [g/min]	Laser On/Off Wait [ms]	Substrate Temperature [°C]
550	12	19.9	400	300

**Table 4 materials-14-07302-t004:** Three-point bending test results.

	Substrate Inconel 625	LENS Cladded Inconel 625	AM Inconel 625
**Bending strength R_g_ [MPa]**	1460 ± 10	1520 ± 35	1535 ± 40

## Data Availability

The data are available in a publicly accessible repository.

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
