# Peer review of "Suitability of Laser Engineered Net Shaping Technology for Inconel 625 Based Parts Repair Process"

_materials, 2021, doi:10.3390/ma14237302_

Round 1

Reviewer 1 Report

This manuscript studies the Inconel 625 laser cladding formed by LENS technology, provides the optimized LENS process parameters, and analyzes the mechanical properties of the laser cladding. The topic should be within the scope of this journal and interesting for the readers. However, the manuscript needs further modification before it can accepted for publication. The authors should correct the manuscript according to the following items:

  1. The topic of the manuscript is to verify that Inconel 625 can be used in LENS cladding technology, and the mechanical properties of the cladding layer are higher than that of the substrate.

From the available literatures, Inconel 625 have been widely used in additive manufacturing and laser repairing, such as ‘Additive manufacturing and hot-fire testing of liquid rocket channel wall nozzles using blown powder directed energy deposition inconel 625 and JBK-75 Alloys’;  ‘https://doi.org/10.1016/j.msea.2015.03.043’; ’ Repairing feasibility of SS416 stainless steel via laser aided additive manufacturing with SS410/Inconel625 powders’.

Can you add more recent progresses in the Introduction and Section 3.1 to let readers   clearly understand the necessity and innovation of your manuscript?

  1. The authors did a lot of works in section 3.1 to analyze the parameter optimization of the LENS process. When laser power is 500 W, the powder feeding speed is 12 RPM, the substrate is heated to 300 ℃, and the three-layer cladding condition, the bonding quality of the substrate and the cladding layer can be improved. This result is very useful. Can you offer the cause of formation of defects between the substrate and the deposited material in Figure 5, and analyze the types of defects? This analysis should be very important for the optimization of process.
  2. The authors use a three-point bending experiment in Section 3.3 to compare the bending strength of three different specimens. It is proved that the flexural strength of the repaired specimen is slightly higher than that of the original material of specimen. This work is meaningful. But the authors should explain the reasons?
  3. I suggest that the author add more research on parameter control for optimizing the LENS laser repairing process.
  4. It is recommended to add a text description in Figure 3

Author Response

Detailed Response to Reviewer Comments

Ms. Ref. No.: materials-1452802

Title: Suitability of Laser Engineered Net Shaping technology for Inconel 625 based parts repair process

Materials Dear Sir or Madame,  

I would like to thank you very much for your letter and the reviewer’s comments on our manuscript (No.: materials-1452802). We appreciate your very valuable comments, that gave us a  chance for revising the manuscript.

We have addressed all of the comments and revised the manuscript accordingly. All of the changes have been highlighted in yellow in the revised manuscript. Detailed responses to the comments are described in the “Response to Reviewers” point by point.

We now resubmit the manuscript for your further consideration for publication in your journal. We sincerely hope this revised manuscript will be finally acceptable for publication. If you have any questions about this manuscript, please do not hesitate to contact me.

Best regards

Izabela Barwinska

On behalf of all co-authors

Institute of Fundamental Technological Research

Polish Academy of Sciences

Reviewer’s Comments:

Reviewer #1:

This manuscript studies the Inconel 625 laser cladding formed by LENS technology, provides the optimized LENS process parameters, and analyzes the mechanical properties of the laser cladding. The topic should be within the scope of this journal and interesting for the readers. However, the manuscript needs further modification before it can accepted for publication. The authors should correct the manuscript according to the following items:

  1. The topic of the manuscript is to verify that Inconel 625 can be used in LENS cladding technology, and the mechanical properties of the cladding layer are higher than that of the substrate.

From the available literatures, Inconel 625 have been widely used in additive manufacturing and laser repairing, such as ‘Additive manufacturing and hot-fire testing of liquid rocket channel wall nozzles using blown powder directed energy deposition Inconel 625 and JBK-75 Alloys’;  ‘https://doi.org/10.1016/j.msea.2015.03.043’; ’ Repairing feasibility of SS416 stainless steel via laser aided additive manufacturing with SS410/Inconel625 powders’.

Can you add more recent progresses in the Introduction and Section 3.1 to let readers   clearly understand the necessity and innovation of your manuscript?

Response: We would like to thank the reviewer for the comment. Corrections were highlighted in the article and detailed response is located below.

First of all, the papers suggested by reviewer were discussed in introduction (lines 69-79).

The laser source based methods, however, offer to overcome the main issues related to the conventionally used methods [12] as well as enables to deposit complex, even high-entropy coatings in order to enhance the mechanical properties of parts working under extreme conditions [13]. Hong et al. [14] used laser metal deposition (LMD) process to produce ultrafine TiC particle reinforced Inconel 625 composite parts, which were characterized by significantly high tensile strength of 1077.3 MPa, yield strength of 659.3 MPa, and elongation of 20.7%. Weng et al. [15] presented a repair process approach by using laser aided additive manufacturing with powder flow rate (LAAM) which was successfully used to deposit SS410 or Inconel625 on SS416 substrate. It was shown, that de-posited clad exhibited no obvious defects and the interface samples exhibited comparative or slightly lower tensile strength in comparison to the SS416 substrate.

The novelty and innovation of the manuscript was however presented in lines 83-108 and 161-164. The main novelty highlighted by authors was that:

  • LENS process is more resource efficient manufacturing technology as less waste is generated in comparison to the subtractive techniques. Additionally, a less amount of material could be used for part repair, that is extremely important in terms of the nickel based alloys application.
  • the main problem of high production cost of parts made of Inconel alloys is overcame by using LENS repair process.

The LENS system enables both, repair of parts and surface modification by the application of protective coating, since the process is characterized by a high accuracy and purity during laser cladding [16-18]. An important advantage of using this system from the met-allurgical point of view is the narrower heat affected zone than that usually obtained in the conventional methods. Other methods deliver large amounts of heat deeply into the material, while in the LENS process the high energy of the laser beam is focused in the relatively small area. In such case, no overheating and subsequent material deformations as well as stress concentrations can be observed [19]. This advantage is very important during repair of the thin walled components. The high power density of laser also pro-motes the application of refractory materials and increases the speed of the process. The resulting laser clads are narrower than in any other processes, and as a consequence, the final part does not require either a rough or finish machining. Additive nature of LENS process makes it a more resource efficient manufacturing technology as less waste is gen-erated in comparison to the subtractive techniques [11-20]. Additionally, a less amount of material could be used for part repair, that is extremely important in terms of the nickel based alloys application.

Nickel alloys, despite their high resistance to high temperature, have the tendency to form the brittle intermetallic phases, which decrease mechanical properties of material [21]. Moreover, the high production cost of parts made of Inconel alloys demands a new techniques, that enable to fabricate complex geometries [22] or repair the broken part without the necessity of their replacement. Therefore, the main aim of this work was to assess the suitability of Laser Engineering Net Shape Technology to repair parts made of the Inconel 625 nickel based superalloy deposited using optimized process parameters. Moreover, since the LENS technology reduces an area of wide heat-affected-zone and do not change the physical characteristics of the deposited material and the substrate, such aspects of this technique were also studied with regard to damaged parts repair.

  1. The authors did a lot of works in section 3.1 to analyze the parameter optimization of the LENS process. When laser power is 500 W, the powder feeding speed is 12 RPM, the substrate is heated to 300 ℃, and the three-layer cladding condition, the bonding quality of the substrate and the cladding layer can be improved. This result is very useful. Can you offer the cause of formation of defects between the substrate and the deposited material in Figure 5, and analyze the types of defects? This analysis should be very important for the optimization of process.

Response: We would like to thank the reviewer for the comment. Additional micrographs were added to Figure 5. Corrections were highlighted in the article (Lines 198 – 206) and detailed response is located below.

The microscopic and tomographic observations revealed, that between the substrate and deposited material many defects can be found including local discontinuities in areas between substrate material and cladding (Figure 5a, d), porosity (Figure 5b) and unmelted particles (Figure 5c). These issues were related to relatively low temperature of substrate material during deposition process in which un-melted or partially melted powder particles could be formed at the interfacial regions between successive layers. Moreover, due to large discontinuities were observed in these regions the powder stream and laser beam were intesivelly scattered at the edges of the pocket.

  1. The authors use a three-point bending experiment in Section 3.3 to compare the bending strength of three different specimens. It is proved that the flexural strength of the repaired specimen is slightly higher than that of the original material of specimen. This work is meaningful. But the authors should explain the reasons?

Response: We would like to thank the reviewer for the comment. Corrections were highlighted in the article (lines 355 -359) and detailed response is located below.

The differences in mechanical response of Inconel 625 alloy in its different states resulted from their initial microstructures and was revealed after observations of bending area. The repaired and additively manufactured Inconel 625 was characterized by columnar structure (Figure 12c-d). Moreover, the laser-based technologies promote the occurrence of Laves phases which are beneficial for the mechanic properties of Inconel alloys [33].

  1. I suggest that the author add more research on parameter control for optimizing the LENS laser repairing process.

Response: We would like to thank the reviewer for the suggestion. The authors optimize the LENS process parameters based on the tomographic and microstructural observations only, thus additional figures were added to Figure 5 and appropriate comments were added in Lines 198 - 206.

  1. It is recommended to add a text description in Figure 3

Response: We would like to thank the reviewer for the comment. The description was added.

Reviewer 2 Report

The manuscript reports the microstructure and properties of Inconel 625 based parts repaired by the Laser Engineered Net Shaping technology, which is worth exploring. However, as far as the current state is concerned, the work still needs improving. Therefore, a mandatory revision of the present manuscript is needed.

  1. Table 1 only provides the chemical composition of nickel based superalloy, but doesnot give the chemical composition of Inconel 625  And there is an obvious error in Table 1 that the unit of chemical composition is wt.% instead of %wt.
  2. It is mentioned in the paper that the processingparameters of powder feed rate remain unchanged, but then it is said that a parameter changed during the test is the powder feed rate.
  3. The laser power is said 500W, but in Table 2 the laser power is 550W?There are many such issues in the article, and the author should check them carefully.
  4. Powder feed rate and powder flow rate have been mentioned several times in the paper. What are the differences?
  5. It is written in Section 3.2 that the width of the heat-affected-zone is 2 μm, but this is not shown in Figure7. Please pay attention to the consistency of the graphic description.
  6. In this paper, manyresults from other papers are described, while less analysis of the present experimental results are made.
  7. The units of microhardness in the paper(HV1) are not consistent with those in the picture (HV0.1). The author should give details of the microhardness test in the paper.
  8. There appear two 3.3 sections.The author should be careful to check the manuscript to avoid similar mistakes.
  9. In thispaper, the author mentioned the additive manufactured material several times, but did not give the specific experimental process.
  10. The author mentioned the hardness of each position- what is the error range, and the corresponding experimental method should be given in the experimental part.
  11. In the introduction, please clearly indicate novelty of the present work.Additionally, the authors are suggested to include a few more new references with closely related topic (HEA/MEA coatings by laser cladding), for example: Applied Surface Science 517 (2020) 146214
  12. English should be well polished before next submitting.

Author Response

Detailed Response to Reviewer Comments

Ms. Ref. No.: materials-1452802

 Title: Suitability of Laser Engineered Net Shaping technology for Inconel 625 based parts repair process

 Materials Dear Sir or Madame,  

I would like to thank you very much for your letter and the reviewer’s comments on our manuscript (No.: materials-1452802). We appreciate your very valuable comments, that gave us a  chance for revising the manuscript.

We have addressed all of the comments and revised the manuscript accordingly. All of the changes have been highlighted in yellow in the revised manuscript. Detailed responses to the comments are described in the “Response to Reviewers” point by point.

We now resubmit the manuscript for your further consideration for publication in your journal. We sincerely hope this revised manuscript will be finally acceptable for publication. If you have any questions about this manuscript, please do not hesitate to contact me.

Best regards

Izabela Barwinska

On behalf of all co-authors

Institute of Fundamental Technological Research

Polish Academy of Sciences

Reviewer’s Comments:

Reviewer #2:

The manuscript reports the microstructure and properties of Inconel 625 based parts repaired by the Laser Engineered Net Shaping technology, which is worth exploring. However, as far as the current state is concerned, the work still needs improving. Therefore, a mandatory revision of the present manuscript is needed.

  1. Table 1 only provides the chemical composition of nickel based superalloy, but doesnot give the chemical composition of Inconel 625  And there is an obvious error in Table 1 that the unit of chemical composition is wt.% instead of %wt.

Response: We would like to thank the reviewer for the comment. Corrections were made according reviewers comment and description of Table 1 was changed.

  1. It is mentioned in the paper that the processing parameters of powder feed rate remain unchanged, but then it is said that a parameter changed during the test is the powder feed rate.

Response: We would like to thank the reviewer for the comment. Powder feed rate did change during the process. Corrections were made in the manuscript (Lines 168-172)

During the laser cladding by Line Biuld Deposition modulus, the test specimens were produced by keeping the process parameters constant at laser power of 550 W, laser spot of 1.5 mm and shielding and carier gas flow rate of 20 LPM and 6 LPM, respectively. The stand-off distance was 13.7 mm. A parameter, that was changed during the test, was the powder flow rate.

  1. The laser power is said 500W, but in Table 2 the laser power is 550W? There are many such issues in the article, and the author should check them carefully.

Response: We would like to thank the reviewer for the comment. Laser power was corrected to 550W.

  1. Powder feed rate and powder flow rate have been mentioned several times in the paper. What are the differences?

Response: We would like to thank the reviewer for the comment. Corrections were highlighted in the article and detailed response is located below. We changed “powder feed rate” to “powder flow rate” throughout the manuscript. The powder feed rate is directly connected with distribution powder method used in LENS system and is specified in revolution per minute (RPM). The powder flow rate is more often used in laser deposition processes. It is independent on used powder distribution method and is specified in gram per minute (g/min).     

  1. It is written in Section 3.2 that the width of the heat-affected-zone is 2 μm, but this is not shown in Figure7. Please pay attention to the consistency of the graphic description.

Response: We would like to thank the reviewer for the comment. The width of heat affected zone is approx. 0.03 mm which is 30 µm. Such width was also observed on micrographs in Figure 7.

  1. In this paper, many results from other papers are described, while less analysis of the present experimental results are made.

Response: We would like to thank the reviewer for the comment. The authors hope, that the improvement made by them will meet the reviewer expectations.

  1. The units of microhardness in the paper(HV1) are not consistent with those in the picture (HV0.1). The author should give details of the microhardness test in the paper.

Response: We would like to thank the reviewer for the comment. Corrections were made and microhardness units were changed in the text to HV0.1.

  1. There appear two 3.3 sections. The author should be careful to check the manuscript to avoid similar mistakes.

Response: We would like to thank the reviewer for the comment. Correction was made according reviewers suggestion.

  1. In this paper, the author mentioned the additive manufactured material several times, but did not give the specific experimental process.

Response: We would like to thank the reviewer for the comment. The optimized LENS process parameters were used to prepare additive manufactured specimens with the same geometry as repaired and wrought one in order to compare their mechanical response (Lines 123-125).

  1. The author mentioned the hardness of each position- what is the error range, and the corresponding experimental method should be given in the experimental part.

Response: We would like to thank the reviewer for the comment. The microhardness of cladding was determined on a ZWICK hardness tester. It was measured every 0.1 mm starting from the edge of the cladding up to its core and repeated in five different cross sections of the repaired specimen.

  1. In the introduction, please clearly indicate novelty of the present work. Additionally, the authors are suggested to include a few more new references with closely related topic (HEA/MEA coatings by laser cladding), for example: Applied Surface Science 517 (2020) 146214

Response: We would like to thank the reviewer for the comment. The additional reference was added to the introduction (Lines 69-72). The novelty and innovation of the manuscript was however presented in lines 83-108 and 161-164. The main novelty highlighted by authors was that:

  • LENS process is more resource efficient manufacturing technology as less waste is generated in comparison to the subtractive techniques. Additionally, a less amount of material could be used for part repair, that is extremely important in terms of the nickel based alloys application.
  • the main problem of high production cost of parts made of Inconel alloys is overcame by using LENS repair process.

The LENS system enables both, repair of parts and surface modification by the application of protective coating, since the process is characterized by a high accuracy and purity during laser cladding [16-18]. An important advantage of using this system from the met-allurgical point of view is the narrower heat affected zone than that usually obtained in the conventional methods. Other methods deliver large amounts of heat deeply into the material, while in the LENS process the high energy of the laser beam is focused in the relatively small area. In such case, no overheating and subsequent material deformations as well as stress concentrations can be observed [19]. This advantage is very important during repair of the thin walled components. The high power density of laser also pro-motes the application of refractory materials and increases the speed of the process. The resulting laser clads are narrower than in any other processes, and as a consequence, the final part does not require either a rough or finish machining. Additive nature of LENS process makes it a more resource efficient manufacturing technology as less waste is gen-erated in comparison to the subtractive techniques [11-20]. Additionally, a less amount of material could be used for part repair, that is extremely important in terms of the nickel based alloys application.

Nickel alloys, despite their high resistance to high temperature, have the tendency to form the brittle intermetallic phases, which decrease mechanical properties of material [21]. Moreover, the high production cost of parts made of Inconel alloys demands a new techniques, that enable to fabricate complex geometries [22] or repair the broken part without the necessity of their replacement. Therefore, the main aim of this work was to assess the suitability of Laser Engineering Net Shape Technology to repair parts made of the Inconel 625 nickel based superalloy deposited using optimized process parameters. Moreover, since the LENS technology reduces an area of wide heat-affected-zone and do not change the physical characteristics of the deposited material and the substrate, such aspects of this technique were also studied with regard to damaged parts repair.

  1. English should be well polished before next submitting.

Response: We would like to thank the reviewer for the comment. The language was polished throughout the manuscript.

Reviewer 3 Report

  1. It is recommended to provide powder feed rate in gm/min rather than RPM.
  2. “power of 500 W and powder feed rate 10 mm/s, flow of working gas on the central nozzle (20 LPM) and nozzle powder (6 LPM)” There are lot of issues with this statement. (a) Units of powder feed rate cannot be in mm/s, (b) The terminology used like working gas and nozzle powder are very uncommon. Try to represent the same in terms of carrier gas flow rate, shielding gas flow rate and shrouding gas flow rate and (c) What is meant by nozzle powder? It is recommended to use ISO/ASTM Terminology.
  3. Also instead of LENS, it recommended to use Laser directed energy deposition (L-DED).
  4. It is very difficult to understand what authors would like to convey by Laser On/Off Wait [ms] and Laser Off/On Shutter Delay [ms]? There are few cases where its value is 1 ms which is not supposed to show any effect. Is authors talking about pulse or modulated mode? If so it should be properly brought out.
  5. What is the spot diameter and stand-off distance used in the present study? What is the relative distance between powder focusing point and substrate surface?
  6. It is interesting to see that laser power was kept constant at 550 W. The quality of deposition in Fig. 5 could have been simply improved by increasing the laser power instead of increasing the substrate temperature by 300 °C.   
  7. Also, as discussed in introduction, increasing the substrate temperature to such high values results in elemental segregations in Inconel 625. Then, why was substrate heating selected?
  8. In table 2, with respect to parameters in S. No 13, parameters in S.No 1 are expected give better metallurgical bonding. However, those results were not represented. It is highly recommend to include the cross-sectional images for all the 13 depositions and discuss the defects with respects to the process parameters.
  9. In Fig. 6 (a) it is recommend to etch the sample and provide the cross-sectional image. In the current format no difference can be observed between the deposited material and the substrate.
  10. What are the parameters used for Micro-Ct tomograph?
  11. “laser powder - 2350 W” It is laser power.
  12. “The homogenous distribution of the based elements observed on the cross-section confirmed, that the LENS system does not lead to the segregation of the alloying elements on the border of cladding and substrate material (Figure 8-9).” Nickel based super alloys like Inconel 625 and 718 suffer from elemental segregation at the grain boundary and not at the interface as highlighted by the authors. It has to be noted that L-DED always results in Laves phases in Inconel 625 whose percentage may be controlled by changing the cooling rates or processing parameters but cannot be made zero. Therefore, the claim of having homogenous distribution cannot be accepted. It is recommend to provide the microstructure and EDS results at hight magnifications.
  13. What was the etchant used in the present study?
  14. In section 3.3, hardness of the clad zone was reported as 275±10 HV. However, in Fig. 10, it can be observed that maximum mean value is close to 268 with error bar below 275. Authors need to revisit these values. Further, no information on how many indentations were collected at each location was provided. Also, how are the error bars obtained? It can be seen that the blue dots do not seem to represent mean value as the error bars on upper side and lower side are not equal.
  15. “The results of the Vickers hardness measurements showed that the hardness of the coating is greater than that of the substrate by approximately 80-100 HV0.3” Once again from Fig. 10 it can be observed that the hardness of clad is approximately 20 HV higher than substrate where as authors claim 80-100 which is nor reflecting in Fig. 10. Further in Fig. 10 load was represented as HV0.1 while in text it is HV0.3. This need to be consistent.
  16. “The increased bending strength of the laser cladded specimens was possibly related to the strengthening of the IN625 alloy due to an influence of the high temperature of the process.” This logic cannot be accepted at all. Authors need to look into the literature as well as their results carefully and bring out a reasonable explanation.
  17. In a 3 point bending test, the lower portion of the material is subjected to tensile load and upper portion to compressive load. Under such condition, claiming a uniform strain is across the material is nor appropriate. In the situations like in current case, tensile specimens need to be obtained in such a way that gauge length consists of both the deposited and substrate material. This will give a better idea of material behaviour. So the results are not convincing.
  18. The results and discussion section in too weak in terms of scientific explanation. Just the results in terms of numbers were represented.

Author Response

Detailed Response to Reviewer Comments

Ms. Ref. No.: materials-1452802

 Title: Suitability of Laser Engineered Net Shaping technology for Inconel 625 based parts repair process

 Materials Dear Sir or Madame,  

I would like to thank you very much for your letter and the reviewer’s comments on our manuscript (No.: materials-1452802). We appreciate your very valuable comments, that gave us a  chance for revising the manuscript.

We have addressed all of the comments and revised the manuscript accordingly. All of the changes have been highlighted in yellow in the revised manuscript. Detailed responses to the comments are described in the “Response to Reviewers” point by point.

We now resubmit the manuscript for your further consideration for publication in your journal. We sincerely hope this revised manuscript will be finally acceptable for publication. If you have any questions about this manuscript, please do not hesitate to contact me.

Best regards

Izabela Barwinska

On behalf of all co-authors

Institute of Fundamental Technological Research

Polish Academy of Sciences

Reviewer’s Comments:

Reviewer #3:

  1. It is recommended to provide powder feed rate in g/min rather than RPM.

Response: We would like to thank the reviewer for the comment. Corrections were made in introduction were error was found and highlighted in the manuscript.

  1. “power of 500 W and powder feed rate 10 mm/s, flow of working gas on the central nozzle (20 LPM) and nozzle powder (6 LPM)” There are lot of issues with this statement. (a) Units of powder feed rate cannot be in mm/s, (b) The terminology used like working gas and nozzle powder are very uncommon. Try to represent the same in terms of carrier gas flow rate, shielding gas flow rate and shrouding gas flow rate and (c) What is meant by nozzle powder? It is recommended to use ISO/ASTM Terminology.

Response: We would like to thank the reviewer for the comment. Corrections were highlighted in the article and detailed response is located below. We improved this paragraph by using proper terminology, i.e. powder flow rate, carrier gas flow rate and shielding gas flow rate. Moreover we changed unit of powder flow rate to g/min (Lines 169-171).   

During the laser cladding by Line Biuld Deposition modulus, the test specimens were produced by keeping the process parameters constant at laser power of 550 W, laser spot of 1.5 mm and shielding and carier gas flow rate of 20 LPM and 6 LPM, respectively. The stand-off distance was 13.7 mm.

  1. Also instead of LENS, it recommended to use Laser directed energy deposition (L-DED).

Response: We would like to thank the reviewer for the comment. Today, directed energy deposition techniques such as, Direct Metal Deposition (DMD), Laser Engineered Net Shaping (LENS™) or Electron Beam Additive Manufacturing (EBAM) are based on similar ideas, but, integrates layered manufacturing concepts to create parts directly from computer-aided design (CAD) data. First commercial effort in directed energy deposition, started with the formation of Aeromet Corporation in 1997, which focused on a laser based directed energy deposition technology.  Commercialization of Sandia National Laboratories led to the development of LENS process in 1998 by Optomec Inc. University of Michigan developed DMD process by POM Group (now DM3D Technology) which brought further thrust into metal additive manufacturing processes. Taking into account the variety of laser based technologies, the authors decided to use Laser Engineered Net Shaping, which precisely defines used deposition method. Moreover, some program modules (e.g. Teach and Learn, Line Build Deposition) and technological parameters (e.g. Laser Off/On Wait) presented in this manuscript are characteristic for LENS method and they are not used anywhere else.

  1. It is very difficult to understand what authors would like to convey by Laser On/Off Wait [ms] and Laser Off/On Shutter Delay [ms]? There are few cases where its value is 1 ms which is not supposed to show any effect. Is authors talking about pulse or modulated mode? If so it should be properly brought out.

Response: We would like to thank the reviewer for the comment. Laser on/off wait mode (specifies how long to delay between depositing each line of material) was used to exclude the extensive deposition of powder near the wall of the repaired pocket. Laser off/on shutter delay time represented an actual time of laser deposition delay between subsequent layers. In fact, laser shutter delay did not exhibit any effects thus it was delated from the table.

  1. What is the spot diameter and stand-off distance used in the present study? What is the relative distance between powder focusing point and substrate surface?

Response: We would like to thank the reviewer for the comment.

The laser spot diameter and stand-off distance (SOD) were 1.5 mm and 13.7 mm, respectively. Unfortunately, the authors don’t know what is relative distance between powder focusing point and substrate surface. The authors adjusted the powder feeder to direct the flow powder only in order to keep it aligned with the laser beam (at 13.7 mm SOD) to achieve optimum deposition.    

  1. It is interesting to see that laser power was kept constant at 550 W. The quality of deposition in Fig. 5 could have been simply improved by increasing the laser power instead of increasing the substrate temperature by 300 °C.   

Response: We would like to thank the reviewer for the comment. When the laser power increase to 600 W, the particles of powder stick to the head of powder feeder and actively block or interrupt the flow of powder during the process.

  1. Also, as discussed in introduction, increasing the substrate temperature to such high values results in elemental segregations in Inconel 625. Then, why was substrate heating selected?

Response: We would like to thank the reviewer for the comment. The segregation in introduction section was related to Inconel 718 and Gas Tungsten Arc Welding (GTAW). The nature of GTAW process is different from LENS as laser used throughout the repair process is depositing the powder in significantly smaller area than GTAW and, what is the most important, do not lead to extensive heating of  repaired material. Additionally, the substrate heating to 300°C do not lead to any microstructural changes of Inconel 625. The only reason to increase the temperature of substrate was to increase the diffusion between it and deposited powder during the process.

  1. In table 2, with respect to parameters in S. No 13, parameters in S.No 1 are expected give better metallurgical bonding. However, those results were not represented. It is highly recommend to include the cross-sectional images for all the 13 depositions and discuss the defects with respects to the process parameters.

Response: We would like to thank the reviewer for the comment. The discontinuities observed in specimens obtained by using different parameters were all comparable as those presented in Figure 5, thus the authors decided to not present them. In order to confirm that statement and answer reviewer question, the tomographic figures for different specimens were presented below.

  1. In Fig. 6 (a) it is recommend to etch the sample and provide the cross-sectional image. In the current format no difference can be observed between the deposited material and the substrate.

Response: We would like to thank the reviewer for the comment. The reason of presenting Figure 6a was to confirm that there is no discontinuities between substrate material and cladding. The microstructural investigations were presented in subsection 3.2.

  1. What are the parameters used for Micro-Ct tomograph?

Response: We would like to thank the reviewer for the comment. Corrections were highlighted in the article and detailed response is located below.

Scanning was carried out using a micro focal X-ray source with 120 kV energy and a focal spot size ~3 µm. The number of projections was 1000 projections over 3600 .

  1. “laser powder - 2350 W” It is laser power.

Response: We would like to thank the reviewer for the comment. Corrections were made and highlighted.

  1. “The homogenous distribution of the based elements observed on the cross-section confirmed, that the LENS system does not lead to the segregation of the alloying elements on the border of cladding and substrate material (Figure 8-9).” Nickel based super alloys like Inconel 625 and 718 suffer from elemental segregation at the grain boundary and not at the interface as highlighted by the authors. It has to be noted that L-DED always results in Laves phases in Inconel 625 whose percentage may be controlled by changing the cooling rates or processing parameters but cannot be made zero. Therefore, the claim of having homogenous distribution cannot be accepted. It is recommend to provide the microstructure and EDS results at hight magnifications.

Response: We would like to thank the reviewer for the comment. Authors provided the microstructure and EDS maps at higher magnification and indeed observed segregation of the elements. Finally we documented obtained results in Figure 8 and added some discussion in Lines274 – 287.

Subsequently, energy-dispersive X-ray spectroscopy (EDS) maps of the laser cladding on substrate was performed. The EDS maps from a scanning microscope with an EDS detec-tor enabled to assess the element distribution of the LENS cladding. The distribution of the based elements observed on the cross-section confirmed, that the LENS system lead to the slight segregation of the alloying elements on the border of cladding and substrate materi-al. Niobium and molybdenum were observed on grain boundaries of cladding (Figure 8). According to Yang et al [27], molybdenum could reduce the solubility of niobium in the dendrite arm and Laves phases. Additionally, the molybdenum addition transforms the Laves phase morphology and decrease the segregation zone around the Laves phase. Moreover, the research of Zhang et al. [28] presents, that the laser repair technology ap-plied for the Inconel 718 alloy components led to elemental precipitation and the occur-rence of the Laves phase. It could be concluded, that the process parameters proposed by the authors in recent study enables to slightly reduce a phase transformation occurred due to the high temperature resulted from the relatively high laser power of 900 W.

  1. What was the etchant used in the present study?

Response: We would like to thank the reviewer for the comment. Corrections were highlighted in the article and detailed response is located below.

The specimens for microstructural observations were etched by using oxalic (90 ml of water +10 ml oxalic acid). (lines114-115)

  1. In section 3.3, hardness of the clad zone was reported as 275±10 HV. However, in Fig. 10, it can be observed that maximum mean value is close to 268 with error bar below 275. Authors need to revisit these values. Further, no information on how many indentations were collected at each location was provided. Also, how are the error bars obtained? It can be seen that the blue dots do not seem to represent mean value as the error bars on upper side and lower side are not equal.

Response: We would like to thank the reviewer for the comment. The improvements were made in the manuscript.

The dots represent the average value from 5 measurements while maximal and minimal value of error bars represents the maximum and minimum value of hardness measurement respectively (Lines 296-297).

The microhardness of cladding was determined on a ZWICK hardness tester. It was measured every 0.1 mm starting from the edge of the cladding up to its core and repeated in five different cross sections of the repaired specimen (Lines 135-137).

  1. “The results of the Vickers hardness measurements showed that the hardness of the coating is greater than that of the substrate by approximately 80-100 HV0.3” Once again from Fig. 10 it can be observed that the hardness of clad is approximately 20 HV higher than substrate where as authors claim 80-100 which is nor reflecting in Fig. 10. Further in Fig. 10 load was represented as HV0.1 while in text it is HV0.3. This need to be consistent.

Response: We would like to thank the reviewer for the comment. The sentences brought by reviewer are corresponding to reference [30], not Figure 10.

  1. “The increased bending strength of the laser cladded specimens was possibly related to the strengthening of the IN625 alloy due to an influence of the high temperature of the process.” This logic cannot be accepted at all. Authors need to look into the literature as well as their results carefully and bring out a reasonable explanation.

Response: We would like to thank the reviewer for the comment. Corrections were highlighted in the article and detailed response is located below.

The differences in mechanical response of Inconel 625 alloy in its different states resulted from their initial microstructures and was revealed after observations of bending area. The repaired and additively manufactured Inconel 625 was characterized by columnar structure (Figure 12c-d). Moreover, the laser-based technologies promote the occurrence of Laves phases which are beneficial for the mechanic properties of Inconel alloys [33].

  1. In a 3 point bending test, the lower portion of the material is subjected to tensile load and upper portion to compressive load. Under such condition, claiming a uniform strain is across the material is nor appropriate. In the situations like in current case, tensile specimens need to be obtained in such a way that gauge length consists of both the deposited and substrate material. This will give a better idea of material behaviour. So the results are not convincing.

Response: We would like to thank the reviewer for the comment. The word uniform was replaced by similar as similar strain distribution in all cases was obtained.

The authors would like to highlight, that uniaxial tensile tests of material in question obtained by using LENS technology were performed and presented in published papers of authors/co-authors of this study:

Microstructure and Properties of Inconel 625 Fabricated Using Two Types of Laser Metal  Deposition Methods Materials 2020, 13, 5050; doi:10.3390/ma13215050

Danielewski, Hubert and Antoszewski, Bogdan. "Properties of Laser Additive Deposited Metallic Powder of Inconel 625" Open Engineering, vol. 10, no. 1, 2020, pp. 484-490. https://doi.org/10.1515/eng-2020-0046

  1. The results and discussion section in too weak in terms of scientific explanation. Just the results in terms of numbers were represented.

Response: We would like to thank the reviewer for the comment. The authors hope, that the improvement made by them will meet the reviewer expectations. The authors do believe that engineering approach of this study will lead to the improvement of repair processes for Inconel based alloys.

Round 2

Reviewer 1 Report

The manuscript  can be accpeted at the present form. The authors already

corrected according to my sugestion.

Reviewer 2 Report

With a satisfactory improvement to the manuscript by the current authors, it is recommended to accept it as its current status.

Reviewer 3 Report

Authors have addressed all the queries of the reviewer and paper in its current form can be accepted.